# Molecularly Imprinted Chitosan-Based Thin Films with Selectivity for Nicotine Derivatives for Application as a Bio-Sensor and Filter

**DOI:** 10.3390/polym13193363

**Published:** 2021-09-30

**Authors:** Obinna Ofoegbu, David Chukwuebuka Ike, Gaber El-Saber Batiha, Hassan Fouad, Roongnapa S. Srichana, Ian Nicholls

**Affiliations:** 1Organic, Polymer, Nano Materials and Molecular Recognition Research Group, Department of Chemistry, Joseph Sarwuan Tarka University, Makurdi 970101, Nigeria; ike.david@uam.edu.ng; 2Department of Pharmacology and Therapeutics, Faculty of Veterinary Medicine, Damanhour University, Damanhour 22511, Egypt; dr_gaber_batiha@vetmed.dmu.edu.eg; 3Applied Medical Science Department, Community College, King Saudi University, P.O. Box 10219, Riyadh 11433, Saudi Arabia; menhfef@ksu.edu.sa; 4Molecular Recognition Materials Research Unit, Department of Pharmaceutical Chemistry, Faculty of Pharmaceutical Sciences, Prince of Songkla University, Songkhla 90112, Thailand; roongnapa.s@psu.ac.th; 5Centre for Biomaterials Chemistry, Linnaeus University, 39782 Kalmar, Sweden; ian.nicholls@inu.se

**Keywords:** chitosan, grafting, methylmethacrylic acid, molecular imprinting, dual templating, thin film, biosensor

## Abstract

This study reports the feasible use of chitosan as a thin film biosensor on the very sensitive quartz crystal micro balance system for detection of blends of multiple templates within a single matrix. The development of chitosan-based thin film materials with selectivity for nicotine derivatives is described. The molecular imprinting of a combination of nicotine derivatives in N-diacryloyl pipiradine-chitosan-methacrylic acid copolymer films on quartz crystal resonators was used to generate thin films with selectivity for nicotine and a range of nicotine analogues, particularly 3-phenylpyridine. The polymers were characterized by spectroscopic and microscopic evaluations; surface area, pore size, pore volume using Breuner-Emmet-Teller method. Temperature characteristics were also studied. The swelling and structure consistency of the Chitosan was achieved by grafting with methylmethacrylic acid and cross-linking with N-diacrylol pipiradine. A blend of 0.002 g (0.04 mmol) of Chitosan, 8.5 μL Methylmethacrylic Acid and 1.0 mg N-diacrylol pipradine (BAP) presented the best blend formulation. Detections were made within a time interval of 99 s, and blend templates were detected at a concentration of 0.5 mM from the Quartz crystal microbalance resonator analysis. The successful crosslinking of the biopolymers ensured successful control of the swelling and agglomeration of the chitosan, giving it the utility potential for use as thin film sensor. This successful crosslinking also created successful dual multiple templating on the chitosan matrix, even for aerosolized templates. The products can be used in environments with temperature ranges between 60 °C and 250 °C.

## 1. Introduction

Despite the promising and evidenced utilization of Chitosan in paper-sizing [1], the food industry [2], pharmaceuticals, Veterinary medicine [3,4] medicine [5,6], separation processes [7,8,9], and energy systems such as fuel cells [10,11], and at different application sizes such as the Nano and micro scales [12,13,14], as well as in the area of molecular imprinting [15,16,17,18], its extensive use in a diversified industrial economy has been limited due to its hydrophilic nature and narrow application temperature tolerance.

The presence of hydroxyl functionality catalyzes its inherent disposition to swell in the presence of water and makes it challenging for thin film studies to be carried out using it as a matrix. In a bid to overcome these challenges, scientists have adopted approaches like copolymerization and grafting in order to stiffen and restrict the swelling potential, amongst other target goals [19,20,21,22,23,24].

Molecularly imprinted fabrications have progressed over the years from uni-template systems with specific single analyte detections [25,26,27] to multi-template systems with selective family analogue detections [28,29,30] and recently uni-template systems with the ability to select multi-analyte protocols [31,32,33]. These systems are functional, however, limitations occur as a result of complexity in the contact environment, the stereo-specificity of available analytes and differences in the pH of the environment. Consequently, researchers have recently delved into the modification of matrices with multi-template binding inclusions [34,35,36]. It is definitely advantageous to employ multi-templated MIPs within an environment that has constituent group analogues or family species so as to simultaneously adsorb the pollutants. This reduces the burden of fabricating numerous matrices for individual specific adsorptions with respect to cost, materials consumption and negative environmental impact.

The research being reported here pertains to the fabrication of a proof-of-concept study of a completely biomaterial-based molecularly imprinted polymer (MIP) product with a multi-template system capable of entrapping more than one pollutant in a given single contact exposure.

For the purpose of achieving the set objective of the present study, copolymerization with methyl acrylic acid as a cofunctional monomer is adopted. The template materials are Nicotine and its structural analogue 3-Phenylpyridine. The use of N-diacryloyl pipiradine/BisAcrylo Pipiradine BAP ensures specific cavity formation for binding of template molecule(s) within the MIP structure [37]. Characterization of samples was done using Quartz Crystal Micro scale (QCM) analysis, Infra-Red (IR) and Ultra Violet- Visible (UV-VIS) spectroscopy, Scanning Electron (SEM) and Transmission Electron Microscopy (TEM), as well as surface area and pore volume determination using the Breuner-Emmet-Teller (BET) method.

## 2. Results and Discussion

### 2.1. Quartz Crystal Microbalance Analysis Results

The best mass density of Chitosan for the MIP architecture was obtained by the experimental variation of quantities of Chitosan and identification of the highest residual vibrational frequency on a QCM resonator while all other inputs were kept constant ESD 1. The obtained value was employed in the formulation as presented in Table 1.

After formulation, the samples were coated on activated resonators and polymerized at 60 °C for 2 h, and their respective frequencies determined using the QCM machine. From Figure 1, a non-zero residual frequency was obtained considering that, both at equilibrium and with subsequent injections of rebinding solutions at different concentrations, the difference in frequency (residual frequency) did not give a zero value. This is in agreement with results obtained by other researchers [38]. The implication is that there was a permanent material deposition on the resonators, which confirms a successful polymerization with template cavities. From the results, a formulation containing 0.04 mmol of Chitosan (chip no 3) (Table 1) was selected because it presented a middle cause in selecting the most suitable amount of Chitosan required for mass deposition. This is also supported by the result from Figure 1 where the relative frequency values of the MIPs were higher than that of the NIP, indicative of the difference in mass-density, a consequence of the template(s) imprinting.

The residual frequencies for all samples were significantly higher than that of the NIP. Interestingly, the frequency value for the blend template occurred in between the respective frequencies of individual templates, with that of Nicotine being lower than the blend and that of 3-Phenylpiridine positioned above that of the blend. At lower concentrations, specifically 0.5 mM, the frequencies of both 3-Phenylpiridine and that of the blend template were approximately the same. This implies that at lower concentrations of detections it may be somewhat difficult to specifically identify 3-Phenylpiridine in a cocktail of close analogues, while that of Nicotine distinctly stands out. Longer rebinding time is also required for detection of 3-Phenylpiridine, as shown from the plot (Figure 2). All other detections were made within the time interval of 99 s except for the highest concentration of 3-Phenylpiridine-template sample, which was done at a much higher detection period. Overall, detections of chitosan-based Nicotine and its analogue template, as well as blends of them, can be made using the QCM resonator at micro trace levels despite the hydrophilic character of Chitosan, which prompts its swelling and astronomical increase in mass density during sensing.

The preliminary trial results demonstrated the functionality of the Chitosan-based MIP thin film on the surface of the chip sensor. The best blend formulation favoured the formulation having 0.002 g of Chitosan, and this was used in the synthesis of a dual-template MIP thin film.

The Sensogram study was done using a Phosphate buffer solution (PBS) at a pH of 7.4 and rebinding template solutions with concentrations of 5 mM, 2 mM, 1 mM, 0.5 mM, 0.2 mM and 0.1 mM contained in the PBS carrier solution. An Attana 100 QCM machine was used for the flow injection analysis (FIA) at a temperature of 22 °C, flow rate of 25, time scale of 50 min, frequency scale of 100 Hz. A frequency offset of 0.0 and injection volume of 100 μL and 2 cycle injections were operated at for the rebinding study. Figure 3 shows the Sensogram chart of the rebinding study with three repeat cycles.

The Sensogram (Figure 2) clearly shows the ability of the fabricated MIP thin film to reproducibly bind the blend template materials as contained in the carrier medium without significant alterations in repeat cycles. Another insight was evident when the concentration of the carrier medium was altered, indicative of an electrostatic influence of a carrier medium on the effective binding of templates by MIP samples (ref). The polymer film prepared without the template molecules or the non-imprinted polymer (NIP) failed to rebind any significant template material because the rebinding by the MIP was actually due to imprinting of the templates, as shown in the Sensogram below (Figure 3).

A selectivity and sensitivity study of the MIP was also carried out using Nicotine, Phenylpyridine, Caffeine and Phenylalanine amide as rebinding molecules alongside the blend template system. The choice of the compounds was predominantly due to their closely related structural and physiochemical properties, as is the case with Caffeine. Nicotine and Phenylpyridine were used so as to confirm the relative sensitivity of the MIP material in the presence of only one of the blend components, and possible changes in preference in the presence of the two but in different concentrations. Their individual molar masses are also closely valued, which helps rule out the possible effects of size or steric preference. Figure 4 shows that the selectivity of the structured MIP for Nicotine was least, below 3-Phenylpiridine, which comes below the blend template sample. This shows the selectivity and sensitivity of the MIP even in the presence of component molecules where they exist individually and not combined. This sensitivity is further confirmed by the non-binding of the Caffeine molecule and the reverse slight adsorption of the Phenylalanine amide molecule.

A plot of the residual frequency against concentration of the templates (5–0.1 mM) clearly shows the selectivity and sensitivity potential of the MIP in the presence of the templates (Figure 5).

Further characterization was done to support the above findings; these include temperature dependent changes as well as the earlier mentioned analysis.

From the electronic supporting document (ESD) 2: FTIR spectra of MIP samples. All components of the polymer blend were made up of similar functional groups, particularly OH, CH_2_, CH_3_, NH_2_; similar peaks were observed in the spectra of both the NIP and very significantly amongst the MIPs. Differences were observed in the individual intensities of transmittance, as shown in Figure 6.

The observed overlaps, disappearances and visibilities occur as a result of the cross-linking and graft polymerization processes. The reported [17] characteristic peak of Chitosan, which is within an average of 1555.2 cm^−1^, is observed in all the samples but with differing percentages of transmittance. These are peaks within the 1550 s cm^−1^. The value for sample 3 is the least, with an average of 83.4, while sample 2 has an average of 85.3, sample 4 91.6, and samples 5 and 1 94.2 and 95.3, respectively (Figure 7).

This is commensurate with the quantities of Chitosan added in the individual sample blends, except for the observed shift in correlation between sample 3 and adjacent sample 4, which may be interpreted as due to the relative degree of grafting that occurred with sample 3, resulting in a reduced level in the intensity of the characteristic peak of Chitosan present. This is indicative of a blend with optimal grafting and crosslinking in the synthesis of the MIP. A further confirmation is seen in the relative degree of intensities with the characteristic peak of C–N functionality.

The presence of characteristic strong and broad band peak ranges from 3700–3200 cm^−1^, which are the extension vibration of the N–H functionality and the O–H vibrational stretching of ungrafted or uncross-linked Chitosan molecule, presents support for the cross-linked grafting of Chitosan and Methacrylic acid in the presence of 1,4-Bis (Acryloyl) piperazine (BAP). It is further observed that a shift towards the asymmetric and symmetric –CH_2_ functionality is evidenced in Figure 8, with increased prominence of peaks within the range of 2920s and 2869s [16,17].

From the FTIR spectra of the template eluted MIP samples (Figure 9), the prominent peaks are as shown in ESD III and the representative functional groups are likewise identified as previously shown in Figure 8.

The individual plot of % transmittance against the wave number for Chitosan, MAA and BAP depicts their characteristic assignments as displayed on the graph (Figure 10). Overlap in transmittances occurs at wavenumbers 954.19, 1695.94, 1293.39, 1435.03, 2174.9 and 3073.13 cm^−1^. Chitosan’s characteristic peak at 3328.51 and MAA’s peaks at 2963.23, 2817.87, 2629.64 and 946.74 cm^−1^ on cross-linking with BAP (Figure 11) give a cross-linked matrix with distinct peaks at 1110.75 (confirmatory secondary alcohol C–O stretch), 1298.98 (primary or secondary OH in-plane bending), 1470.43 (carboxylate functionality from carboxylic acid of MAA after cross-linking with Chitosan in the presence of BAP), 1649.35 (primary amine N-H bend), 2126.45 and 3324.79 cm^−1^ (normal polymeric OH stretch and N-H stretch of primary amine).

There are two key factors when identifying the vibrational mode of samples using a QCM, namely, whether the superimposable characteristics of residual vibrations form a mirror image, and whether the dissipation value is greater than 2 × 10^−6^. Where the dissipation value is greater than stated and there is no mirror image of the residual vibrational mode, then the material is said to be viscoelastic; when the reverse is the case, it is said to be elastic [21]. Figure 13 shows the residual vibrational frequencies of the samples of MIPs that were synthesized; it shows a non-mirror image of the residual vibrational frequencies, and dissipation values were above those stated. This affirms that the MIPs exhibit viscoelastic characteristics during the activity period. This is also attributable to the hydrophilic property of Chitosan, which causes a degree of swelling and elasticity.

It is observed that due to blending and interaction with the polymer matrices, the template molecule with initial dual peak absorbance converged to only one significant peak with reduced absorbance maxima. This phenomenon is also observed with the NIP, but with greatly reduced significance considering the fact that there is still dual peak spectra absorbance as a low-positioned elbow in the wavelength range of 250 and 300 nm. The MIP samples show only one prominent peak with a wavelength between 200 and 250 nm. The single prominent peak is a result of the appearance of the templates at an overlapping wavelength and absorbance maximum due to their very closely related electronic π − π * and n − π * configurations.

### 2.2. Temperature Induced Changes of Polymer Films

When subjected to heating in a Bibby Stuart Scientific melting point SMP1 apparatus with a power capacity of 50 W, voltage of 230 V and heating rate of 1 °C/4.9 s, physio-chemical changes were observed with respect to material stability (ESD Table 2). The NIP samples shrank within an average temperature range of 82–98 °C. This is due to evaporative instances of water molecules with weak hydrogen bonds contained in the sample. Samples with only Nicotine showed no visible signs of shrinking, but decomposed at temperature ranges between 100 °C and 300 °C for samples with highest content of Chitosan. This implies that the higher the amount of Chitosan with effective copolymerization, the more stable the product and consequently, the higher the decomposition temperature. The samples with only 3-Phenylpiridine had a higher decomposition temperature but a close shorter-range difference between component blend samples, starting from 171 °C to 240 °C for samples with the highest amount of Chitosan. These samples did not shrink but rather became translucent before decomposition. This is an indication of a level of sustained stability before decomposition for samples having the phenyl group substituent, probably due to the aromaticity of the group. The polymer samples with the blend of templates exhibited both the shrinking and the translucency but in reversed order, becoming translucent before shrinking and then decomposing. This phenomenon was observed from a temperature of 140 °C up to 220 °C. This indicates the effect and evidence of blending between the two templates. This characteristic reflects the observed trend in ESD IV, where the MIP with the blend template had resonant frequency in-between the two individual templates, showing a sort of compensational compromise obtained from blending the templates. This response can be used to identify blends of templates in environments with multi-template molecules.

### 2.3. Scanning Electron Microscopy (SEM)

Field emission scanning electron microscopy using a JEOL Field Emission Scanning Electron Microscope (FESEM) was used to characterize the surface morphology of representative samples of both the MIPs and the NIP; the results are presented as Figure 12, Figure 13 and Figure 14.

The SEM micrographs present the representative samples as being spherical in their morphology. The MIPs (Figure 12 and Figure 13) are observed to have cavities with a linked hierarchical type of network, while the NIP (Figure 14) does not possess such attributes. This is due to the non-templating of the polymer, which ensured that cavities as obtainable with the MIPs were not created. The particles are mesoporous and clustered, presenting a mesh-like framework which is a disadvantage for typical SPE application but acceptable for sequestration of very closely clustered materials as are obtained with fluid cocktails such as smoke streams. The sizes are 200 nm for the MIPs and 100 nm for the NIP. This difference in size easily implicates the templating and non-templating activities carried out during polymerization. The respective surface area (26.455, 76.635 and 5.339), pore volume (0.067, 0.219 and 0.010) and pore diameter (3.411, 3.411 and 4.302) values for 3-Phenylpiridine, Nicotine-3-Phenylpiridine and the non-templated, respectively (Table 2), align with the observed presentation except for the pore diameter of the NIP which is more than that of the MIPs. The MIPs are polydisperse as indicated by their different aggregate sizes [17,18,19,20,21], and this is in tandem with the micrographs of the TEM.

### 2.4. Transmission Electron Microscopy (TEM) Analysis

The analysis was carried out using a JEOL, JEM-2010 Electron Microscope.

### 2.5. Surface Area, Pore Size, Diameter and Volume Analysis, Brunauer-Emmett-Teller (BET)

BET results were obtained from Quanta chrome Nova Win, NOVA Quanta chrome Instruments with software version 11.03. The conditions prevailing as at the period of analysis were:Adsorbate: NitrogenAdsorbate Temperature: 77.350 KOutgas Time: 2 hOutgas Temp: 100 °CAnalysis gas: NitrogenBath Temp: 273 KPressure Tolerance:0.100/0.100 (ads/des)Equilibrium time: 60/60 s (ads/des)Equilibrium timeout: 240/240 s (ads/des)


The samples present a typical composite type IVa and type II isotherm, exhibiting typical features of a type II isotherm including unrestricted monolayer-multilayer adsorption with the characteristic “sharp knee” [21] which represents the completion of monolayer coverage. The curve then extends, in a less distinctive curve relative to the “sharp knee”. This is proof of a substantial degree of overlap between the monolayer coverage and the beginning of multilayer adsorption. This phenomenon combines with type IV isotherm-characteristic conical and cylindrical mesopores that are closed at the tapered end, as seen in the results of the TEM analysis (Figure 15, Figure 16 and Figure 17). The tapered ends of the pores are presented as converging cavities. Since the pores’ widths are on average wider than 4 nm (Figure 18, Figure 19, Figure 20, Figure 21 and Figure 22), the adsorption occurred via capillary condensation, hysteresis was undergone as part of the adsorption mechanism [17], and the samples exhibited H3 type hysteresis with characteristic adsorption branching phenomenon of the Type II isotherm (“sharp knee”); this corroborates the earlier deduction of the samples being type IV isotherms. Another characteristic is the branching at the lower limit of desorption being normally cited at the cavitation induced pressure point (p/p0). From the hysteresis loop which occurred at relatively high pressure, the samples are distinctively mesoporous, in agreement with reports by other researchers [15]. Values of the surface areas, pore volumes and pore diameters were all below 100 m^2^ g^−1^ for surface, 2 ccg^−1^ for pore volume and 4 nm for pore diameter (Table 2). This points to the MIPs and NIPs being nanoparticle-sized. This aligns with the submission from [15,21]. The Temographs show non-uniformity in the dimensions of the cavities for both the MIPs and NIP. This can be linked to influence from weak template-monomer interactions and with the peak centers being above 4 nm; it is strongly agreeable that the pore dimensions were generated during polymerization via template-monomer interactions rather than porogen/solvent influences [17,21].

## 3. Experimental

### 3.1. Reagents and Materials

Low molecular weight Chitosan was procured from Sigma Aldrich Ltd. Darmstadt Germany.; Nicotine from Fisher scientific GTF AB Goteborg, Sweden. Triethylamine (TEA) was obtained in its analar grade from MERCK Ltd. Darmstadt Germany. Methacrylic acid (MAA), 1,4-Bis (Acryloyl) piperazine (BAP), Ammonium peroxydisulfate, Toluene, Acetone, Tetrahydrofuran (THF), Hydrogen Peroxide, Sulphuric Acid and were all of Analar grade and procured from SIGMA-ALDRICH Ltd. Darmstadt Germany.

### 3.2. Apparatus and Measurements

An Attana 100 QCM machine was used to carry out the injections and recording of the resonant frequency changes that occurred as a result of template introductions and elution. A CARY 630 FTIR spectrometer by Agilent Technologies was used to confirm the presence of functional groups within the polymer compounds and effective polymerization/cross linking/templating. A Scanning Electron Microscope (SEM) (Leo 1550 Gemini instrument furnished with a field emission electron gun in the high vacuum mode), was employed in the characterization of the surface morphology. Transmission Electron Microscopy (TEM) analysis was carried out using a JEOL, JEM-2010 operating at 160 kV. The samples were suspended in Phosphate Buffer Solution (pH 4) deposited drop-wise and evaporated on 200 mesh copper grids. The Electron Microscope aided the characterization of the cavities formed after polymerization. TEM enhanced the aggregate units as distinctly resolved and analyzed individual units. The surface area, pore size and pore volume analysis using the BET Method was done using Quanta chrome Instrument, with Nova Win, NOVA Quanta chrome Instruments, version 11.03. data analysis software.

### 3.3. Cleaning of Chip Surface

The preparative step of cleaning the quartz crystal chip surface was done by combining an adapted method [38], where SiO_2_-coated crystals were washed in piranha solution (70% H_2_SO_4_ and 30% H_2_O_2_) by gentle swirling of the chips in the solution for 2 min. A liberal amount of double distilled/ultrapure water was then used in rinsing the chips before neutralization in a 0.1 M NaOH solution, aided by sonication for 20 min followed by drying in a stream of Nitrogen gas. The pre-cleaned chips were sonicated in Acetone then Tetrahydrofuran and finally Toluene for 20 min each, respectively. They were finally dried using Nitrogen gas and kept in Falcone tubes for the next stage of silanization.

### 3.4. Functionalization of Chip’s Surface (Silanization)

Silanization was carried out on the cleaned surfaces of the chips by immersing the chips in a solution containing 7.2 μL of 3-(Trimethoxysilyl) propyl methacrylate (silane), 0.72 μL of Triethylamine (TEA) and 360 μL of Toluene for not less than 24 h; they were then wrapped with Aluminum foil and kept in a cupboard. At the end of the contact period, the chips were sequentially rock-washed for 30 min each in Toluene, THF and Acetone, then allowed to dry in a stream of Nitrogen gas.

### 3.5. MIP and NIP Preparation

Imprinted and non-imprinted polymer samples were prepared using a blend ratio of 1.6:12:55:1:1.26 for solvent, functional monomer cross linker, template, and initiator, respectively. The blend was vortexed for 30 s before a 0.5 μL volume was introduced onto the surface of the functionalized crystal chip. The chips containing the pre-polymerized matrix were exposed to thermal activation (60 °C) by surface-induced free radical polymerization process. The MIP was prepared with the template materials Nicotine (**MIP Product A**), 3-Phenylpiridine (**MIP Product B**) and a 50:50 blend of both templates (**MIP Product C**); while the NIP was prepared without templates. Table 1 shows the blend formulations for the samples prepared.



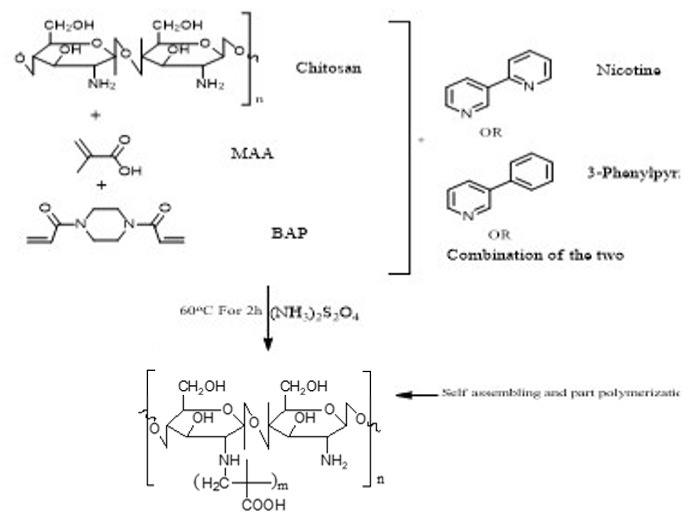





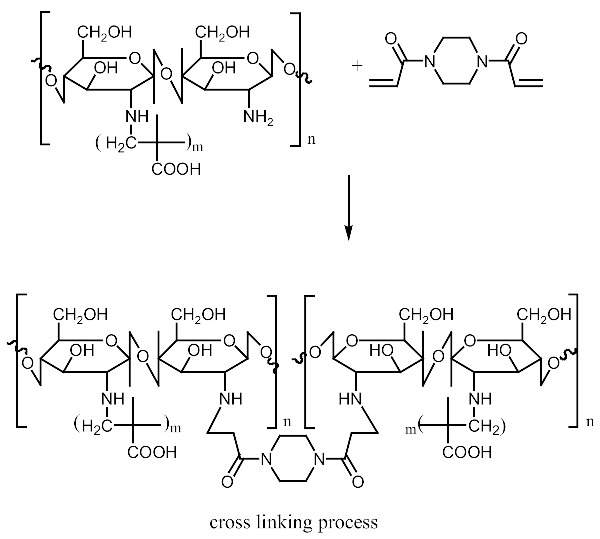



**MIP Product A, B and C**.



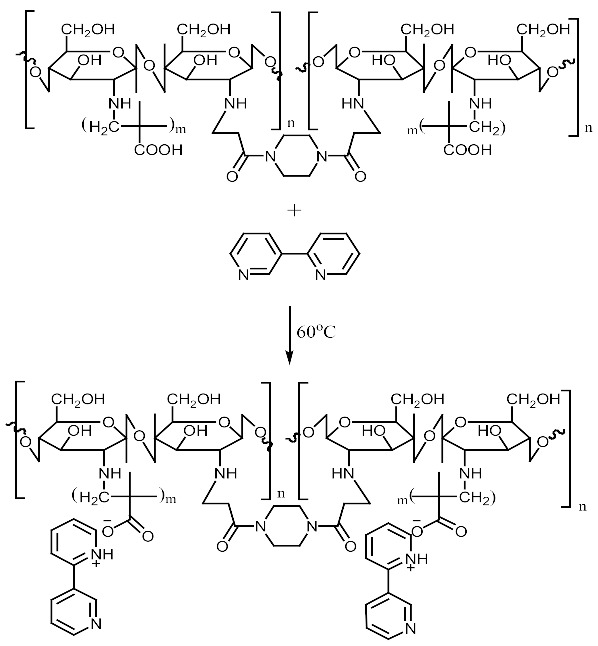



**MIP Product A**.



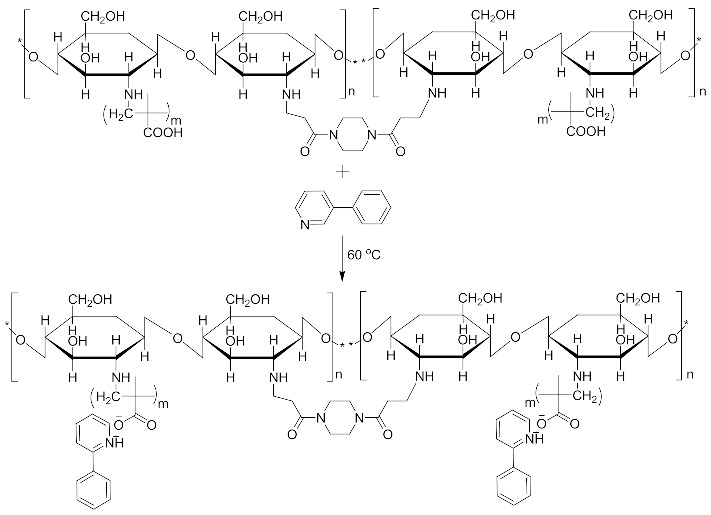



**MIP Product B**.



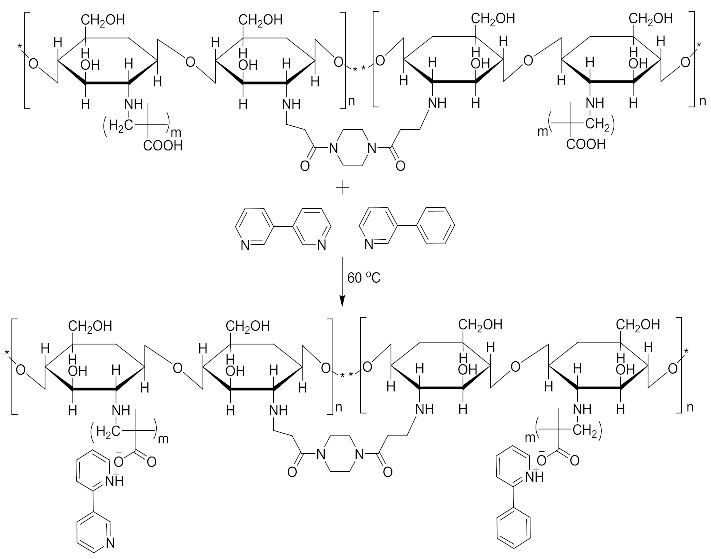



**MIP Product C**.

### 3.6. Elution of Template Materials

At the end of the polymerization, the polymer films on the surface of the chips were washed first with 1 mL Acetone, then 1 mL of Acetonitrile (polymerization solvent for sample 2), Methanol (for sample 4) and finally with 0.01 M NaOH for both the MIPs and the NIP samples. This was done in two washing cycles, for 10 min each using the rock-washing equipment. This treatment eluted the template materials from the polymer films. The MIPs and NIP were then kept for testing and analysis using the Attana QCM machine.

### 3.7. Instrumentation

FTIR spectroscopy allows the possibility of quantitatively mining evidence with respect to the arrangement of the synthesized NIP or MIP. Considering the cocktail of chemical fluxes within the prepared matrix as a result of interactions between the additives, it becomes very necessary to use this analytical tool to characterize samples based on the functional groups present or absent and the interplays that occur during the polymerization, template elution and/or rebinding. The spectrum from an IR analysis facilitates the examination of the type of interaction(s) that took place as a result of the polymerization reaction during MIP synthesis. It particularly gives clear insight into the bond types that exist, such as non-covalent hydrogen bonding or other weak bonding phenomena. From an interpretation of the bonding type, plausible mechanisms of the interaction between template and matrix are also understood and this also directs the scope of specific interactions and preferences for the monomer-cross linker interaction and monomer-template relationship, all of which help in predicting the rebinding ability of the complex matrix. [39]. Functional groups’ presence or absence have been identified [40] and used by researchers to draw conclusions on the effective elution of templates. [41]

Shunli et al., 2013 observed strong bands at 3441.76 cm^−1^ matching the OH stretching vibration, bands at 3441.76 cm^−1^, 1732.5 cm^−1^ and 1259.65 cm^−1^ of carboxyl functionality, as well as absorbance at 1638.63 cm^−1^ consigned to stretching of C–C bond, pinpointing differences between the NIP and MIP samples. This can be authenticated even in the presence or absence of the template material [42]. FTIR has been used to identify harmful components in pesticides [43] and other organic substances. UV-VIS spectroscopy provides information on the presence and degree of template material with respect to the MIP matrix. This is done by noting the evidenced absorbance from the NIP and MIP, or in some comparative studies, differences in absorbance values. This analysis also gives insight into the binding capacities of matrix and template. Reports have presented cases where it was used to characterize substances like benzotriazole [44], tetracycline [45], b-estradiol, estradiol benzoate, ethynyl estradiol etc. [46], as well as the extent to which binding was feasible.

The surface morphology of synthesized polymer matrix influences the chemical, binding and thermal capacities of analyte-matrix system, which has a direct relationship with binding specificity within the polymer matrix [47,48]. This then necessitates the study and characterization of synthesized NIPs and MIPs using SEM. Light microscopy verifies the fundamental physical reliability of the matrix’s globules while SEM confirms the image macrospores [49].

Apart from the fact that QCM operates at sensitivities up to the nanogram level, it also has the ability to detect elusive variations within an environment which may occur as a result of mass-volume-viscosity in a medium. It detects variations within the rigid material as these affect the viscoelastic properties. Moreover, the surface free energy parameter is characteristically accounted for when using this technology. Consequently, QCM presents a very cost effective, fast and efficient tool for spot confirmation of the feasibility of the synthesis of molecularly imprinted materials. This technique has contributed to the clarification of several inherent properties of biological systems. This is achieved by studying the direct correlation between change in frequency (Δf) and mass per volume quantity (Δm) [50] by the application of its sensitivity at the sub mono layer level. Despite its drawbacks [51] such as violation of the Sauerbrey assumption of adsorption occurring at the rigidity state of samples, QCM has found application in many areas of biomaterials science [52,53,54].

## 4. Conclusions

From the results obtained, a matrix based on Chitosan copolymerized with MAA was synthesised and this was simultaneously imprinted with two template materials, Nicotine and an analogue. QCM analysis established the viscoelastic properties of the polymer thin film, as well as the feasibility of having a blend of more than one template within a singular matrix. These MIPs can be used in environments at temperatures above 60 °C but below 250 °C, as shown by the of temperature effect results obtained using the melting point apparatus. Obtained experimental data also showed that chitosan solutions of very low concentrations (0.04 mMole) are suitable for thin film operations and exhibit potential for multi-templating with target materials. Detection of the template materials is feasible even at micro-trace level of 0.1 mMoles and a chitosan weight of 0.002 g.

The selectivity and sensitivity of the prepared MIPs were authenticated by their response to the presence of caffeine and phenylalanine amide within the same environment, where the templated MIPs selectively removed the vinyl alanine amide, in preference to the closely related caffeine. The SEM result authenticated the spherical nature of the prepared MIPs, while the TEM results confirmed the cavity integrity of as little as 27.19, 51.21 and 150.93 nm, which created the ability of the prepared MIPs to sequester toxicant even at trace micro levels.

## Figures and Tables

**Figure 1 polymers-13-03363-f001:**
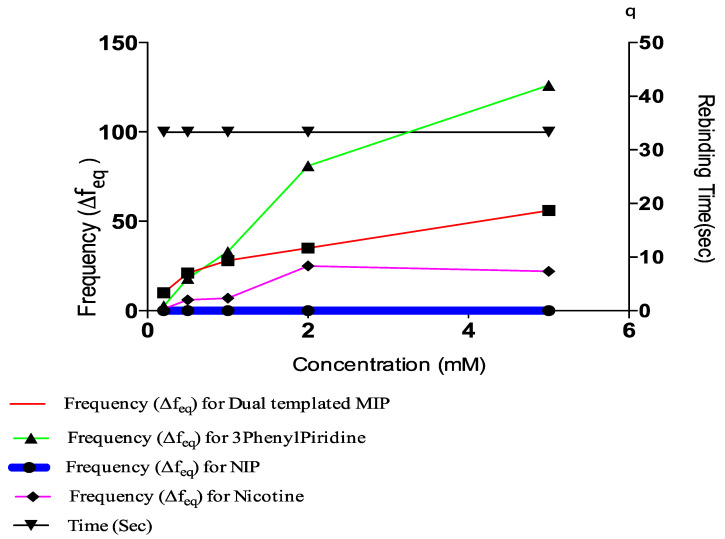
Comparative plot of concentration, rebinding time and frequency.

**Figure 2 polymers-13-03363-f002:**
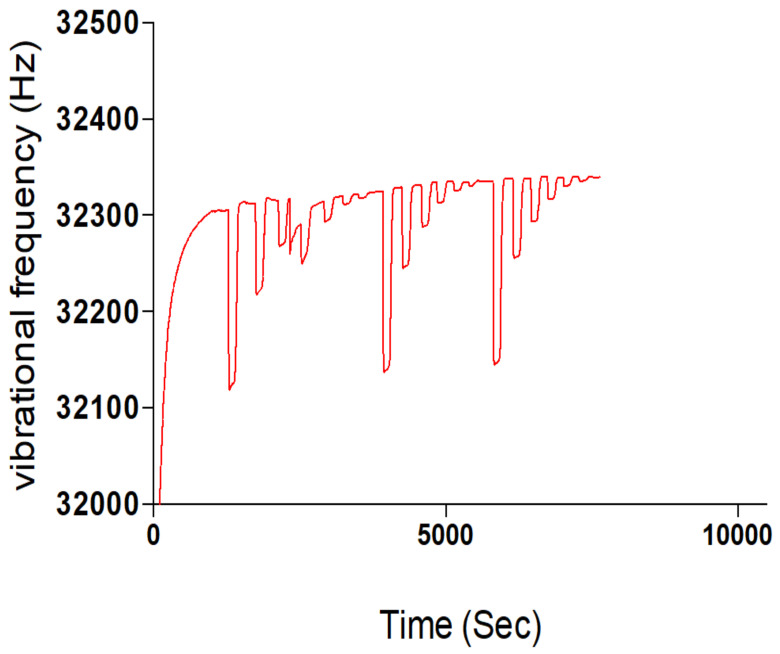
Template rebinding potential of synthesized Chitosan-Nicotine-3-Phenylpiridine MIP.

**Figure 3 polymers-13-03363-f003:**
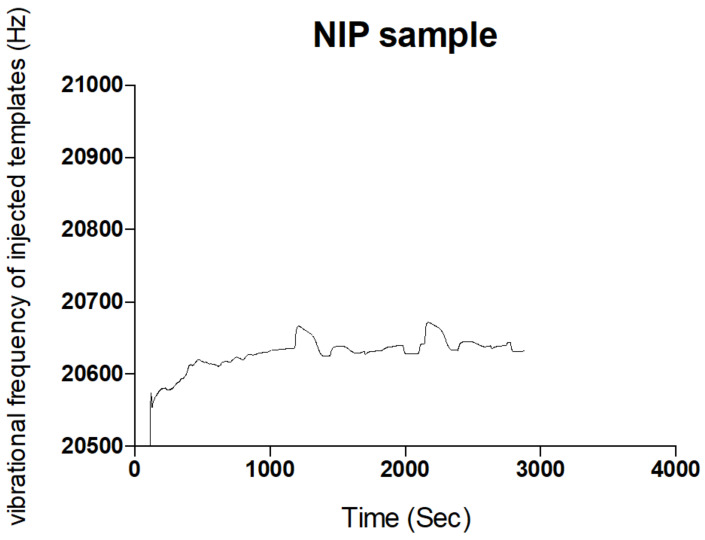
Sensogram of non-rebinding activity of NIP sample from the fluid injection analysis.

**Figure 4 polymers-13-03363-f004:**
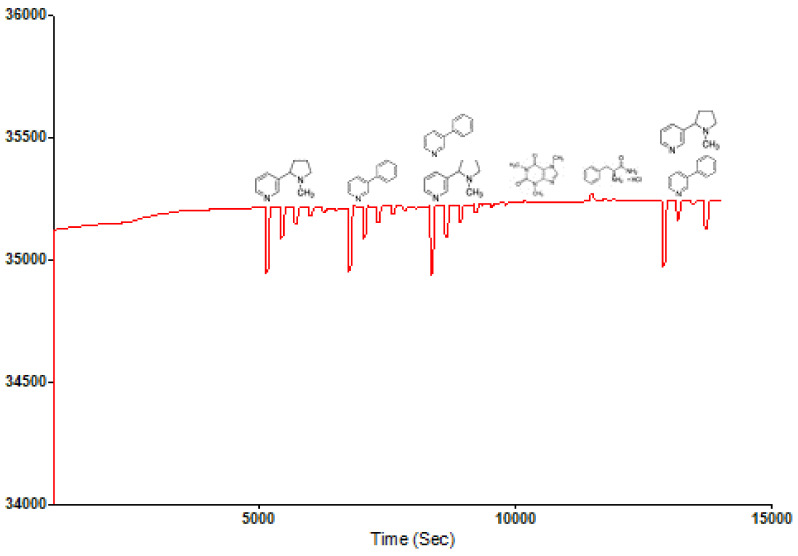
Selectivity and sensitivity Sensogram of Nicotine-3-Phenylpiridine chitosan-based MIP.

**Figure 5 polymers-13-03363-f005:**
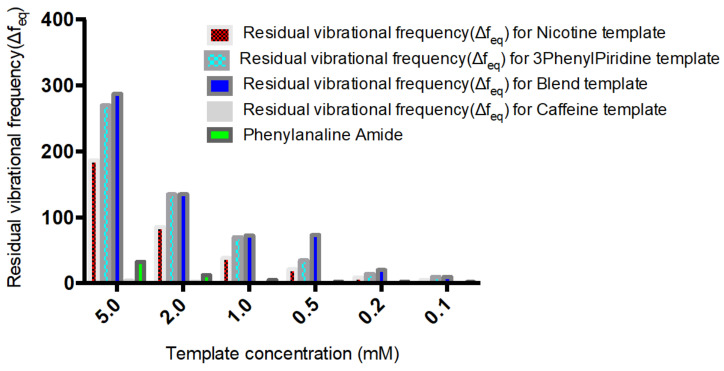
Sensitivity and selectivity of Nicotine, 3-Phenylpiridine, Nicotine-3-Phenylpiridine and Caffeine chitosan-based MIP.

**Figure 6 polymers-13-03363-f006:**
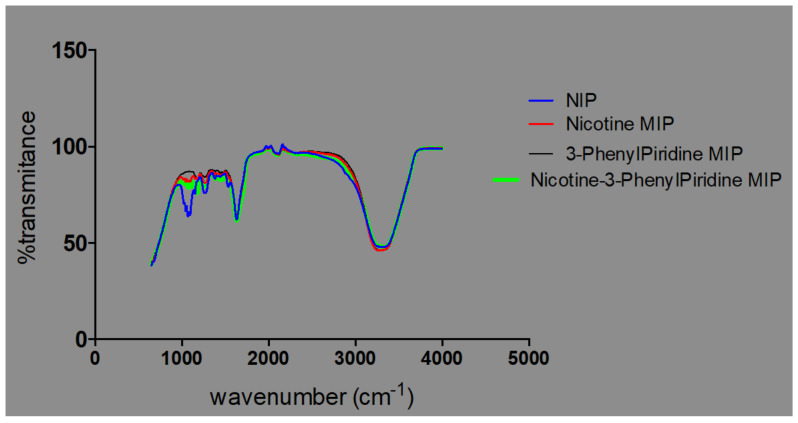
Plot of transmittances against % intensities for NIP- and MIP-containing templates.

**Figure 7 polymers-13-03363-f007:**
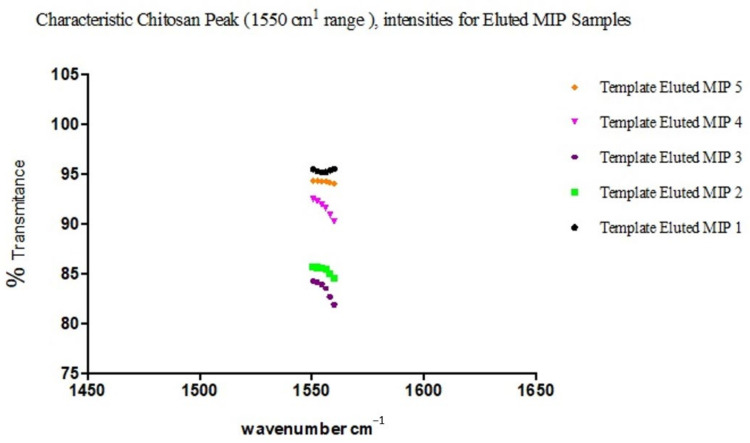
Characteristic peak intensities of Chitosan (1550 cm^−1^) for the different eluted MIP samples.

**Figure 8 polymers-13-03363-f008:**
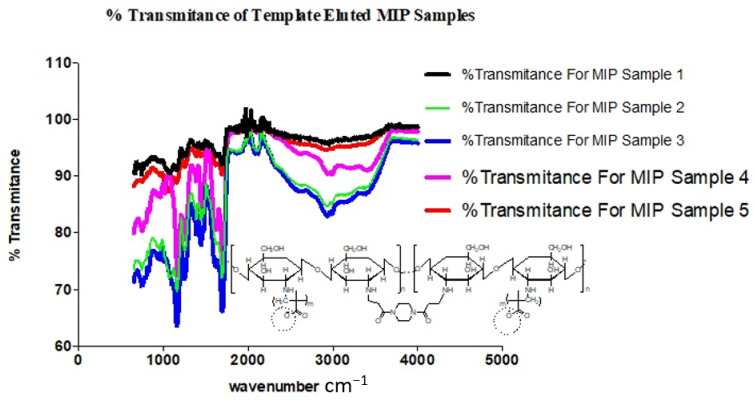
Spectra chart of % transmittance for template-eluted MIP samples.

**Figure 9 polymers-13-03363-f009:**
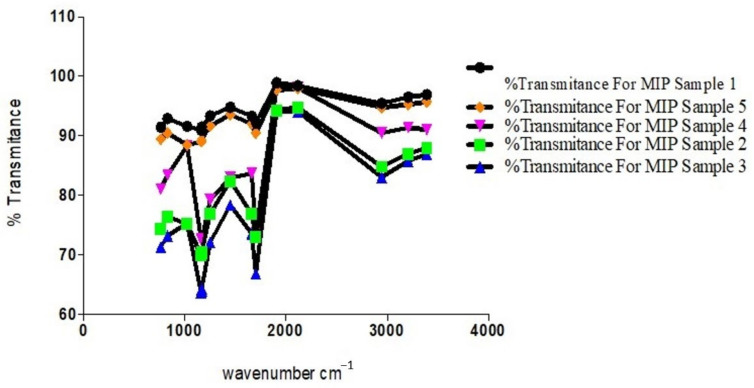
Plot of transmittance against wavenumbers of prominent peaks of MIP samples.

**Figure 10 polymers-13-03363-f010:**
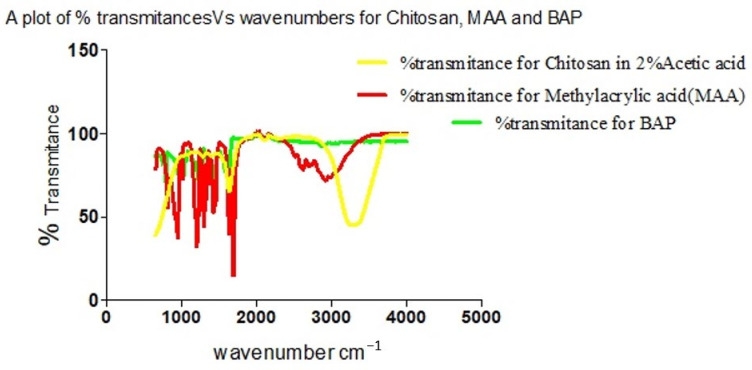
Overlay of spectra of % transmittances against wavenumbers for Chitosan, MAA and BAP.

**Figure 11 polymers-13-03363-f011:**
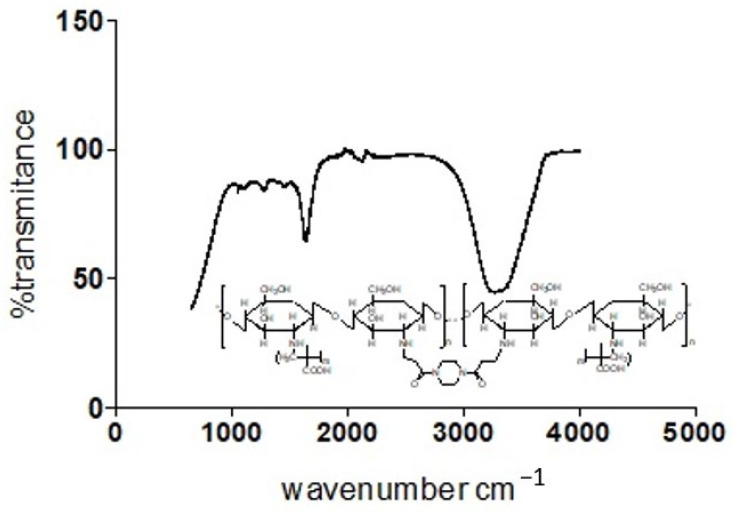
Plot of transmittance against wave number for Chitosan, MAA and BAP.

**Figure 12 polymers-13-03363-f012:**
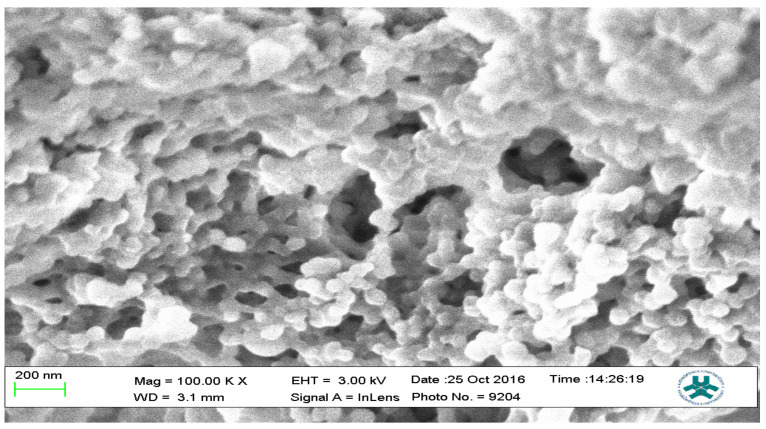
SEM micrograph of the 3-Phenylpiridine MIP.

**Figure 13 polymers-13-03363-f013:**
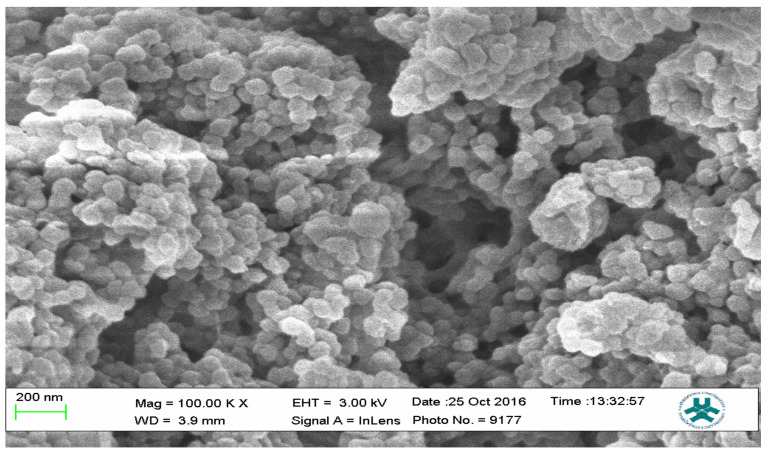
SEM micrograph of Nicotine-3-Phenylpiridine MIP.

**Figure 14 polymers-13-03363-f014:**
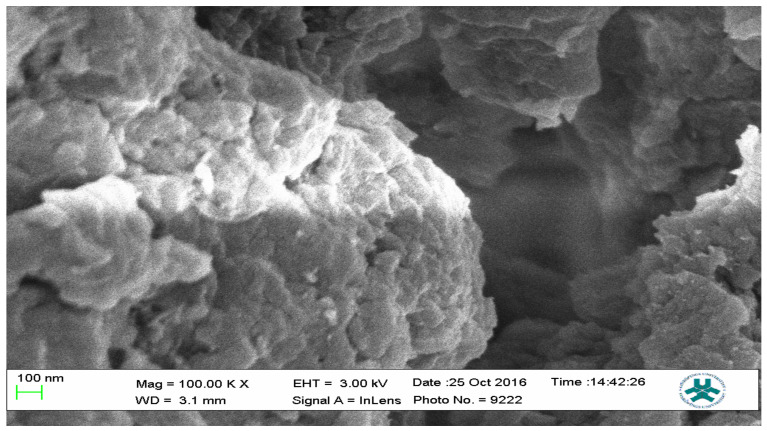
SEM micrograph of the NIP sample.

**Figure 15 polymers-13-03363-f015:**
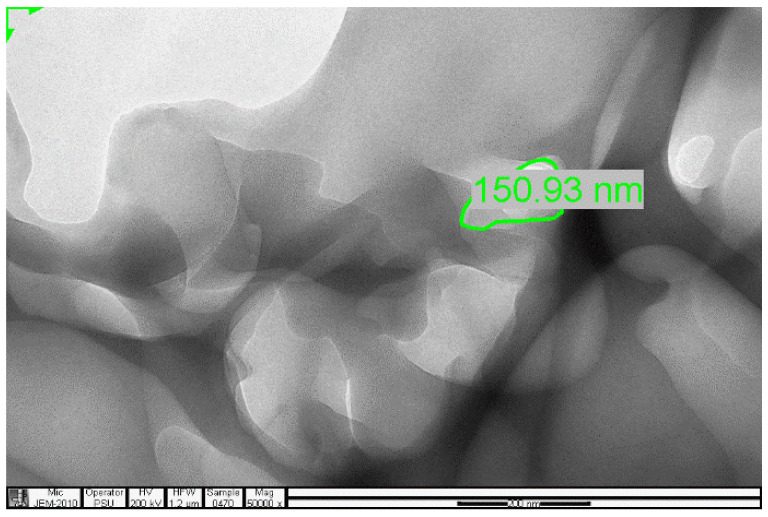
TEM micrograph of the 3-Phenylpiridine MIP.

**Figure 16 polymers-13-03363-f016:**
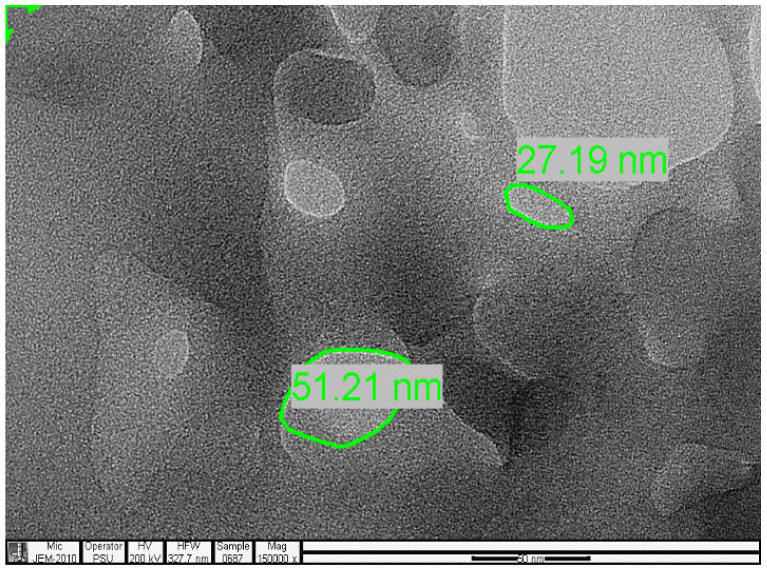
TEM micrograph of the Nicotine 3-Phenylpiridine MIP.

**Figure 17 polymers-13-03363-f017:**
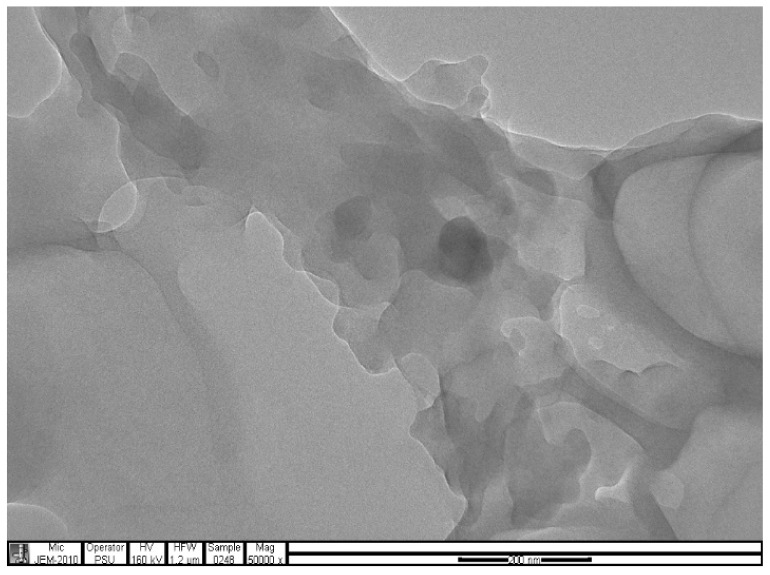
TEM micrograph of the NIP sample.

**Figure 18 polymers-13-03363-f018:**
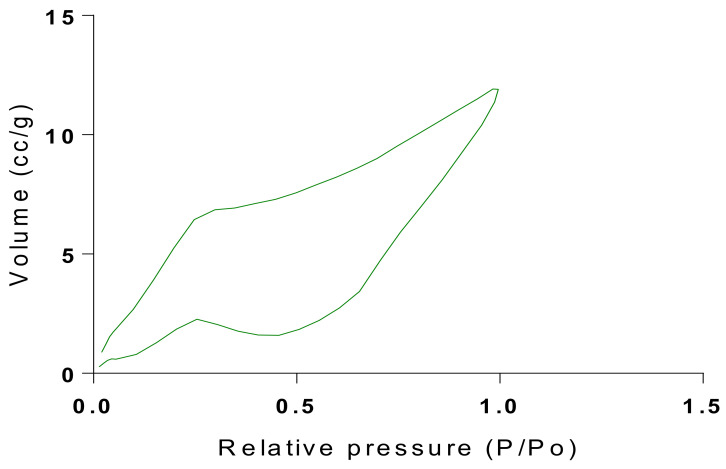
Isotherm of Chitosan-MAA-BAP-3Phenylpridine.

**Figure 19 polymers-13-03363-f019:**
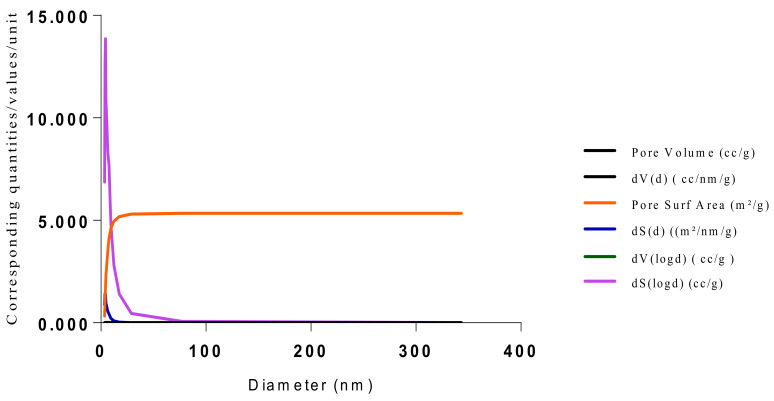
BJH plot for Chitosan-MAA-BAP-3Phenylpridine MIP.

**Figure 20 polymers-13-03363-f020:**
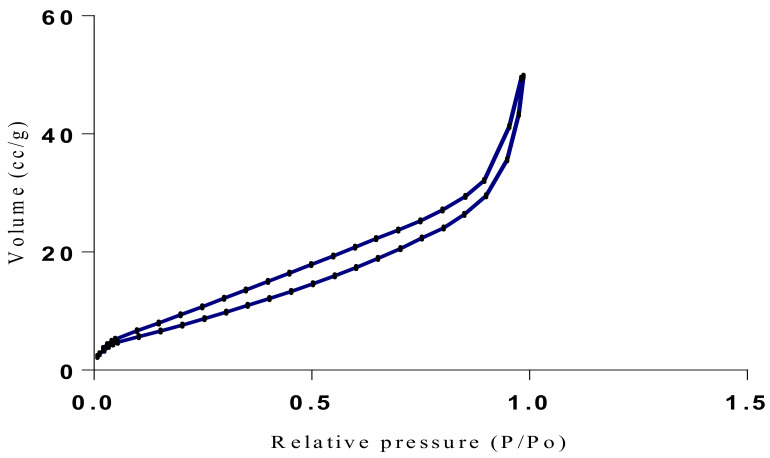
Isotherm of Chitosan-MAA-BAP-Nicotine-3Phenylpridine (MIP).

**Figure 21 polymers-13-03363-f021:**
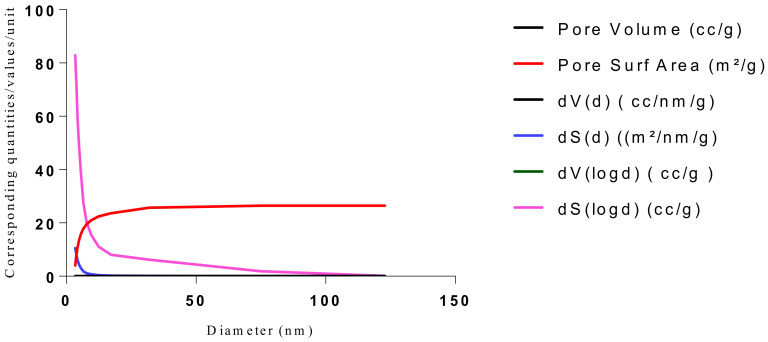
BJH plot for Chitosan-MAA-BAP-Nicotine-3Phenylpridine MIP.

**Figure 22 polymers-13-03363-f022:**
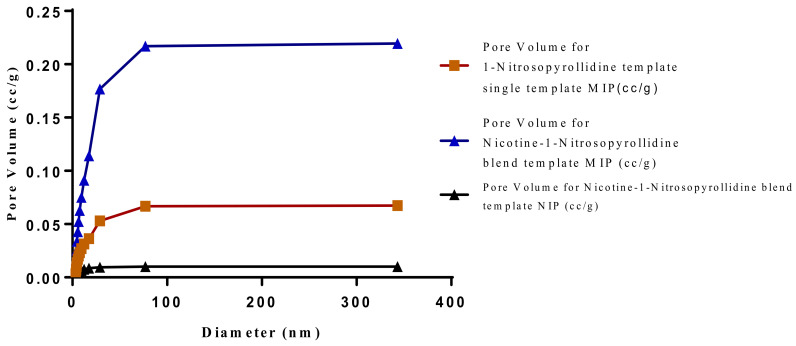
Pore diameter vs. Pore volume for Chitosan-MAA-BAP-3-Phenylpiridine, blend with Nicotine, and NIP.

**Table 1 polymers-13-03363-t001:** Mole = 1 g of Chitosan in 10 mL 2% Acetic acid solution.

Input Materials	Quantities				
	Sample 1	Sample 2	Sample 3	Sample 4	Sample 5
Chitosan (mmol/L)	0.12	0.08	0.04	0.02	0.16
Methacrylic Acid (μL)	8.5	8.5	8.5	8.5	8.5
BAP (mg)	1	1	1	1	1
Template (μL)	7	7	7	7	7
Ammonium persulphate (mg)	0.0001	0.0001	0.0001	0.0001	0.0001

**Table 2 polymers-13-03363-t002:** Summary of BJH desorption result.

Sample	Surface Area (m^2^ g^−1^)	Pore Volume (ccg^−1^)	Pore Diameter (nm)
3-Phenylpiridine MIP	26.455	0.067	3.411
Nicotine-3-Phenylpiridine (MIP)	76.635	0.219	3.411
NIP	5.339	0.010	4.302

## Data Availability

Not applicable.

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
