# Peer review of "Molecularly Imprinted Chitosan-Based Thin Films with Selectivity for Nicotine Derivatives for Application as a Bio-Sensor and Filter"

_polymers, 2021, doi:10.3390/polym13193363_

Round 1
Reviewer 1 Report
The manuscript (polymers-1360217) entitled “Molecularly Imprinted Chitosan-based Thin Films with Selectivity for Nicotine Derivatives for application as a bio-sensor and filter” presents the feasible use of chitosan in thin film applications and in the detection of blends of multiple templates within a single matrix. The products can be used in environments with temperature range between 60 oC and 250 oC. However, the synthesized product is not novel. It can be considered for publication after revision subsequent to addressing the following comments:
- The novelty and originality of this study should be strengthened.
- The author should be revised the Conclusion with proper key results.
- Figures 6 and 7 presented very poorly, need to present properly. Subtitle of Figure 8 need to use correctly. Moreover, Subtitle of Figure 22 is not clear, need to present clearly.
- Almost all cited references are very old. The author must replace the old cited references with latest references (2019, 2020, 2021).
Author Response
Thanks for the thorough review. All correction have been made and highlighted yellow on the manuscript.
Responses to other issues have made on the attachment.

Reviewer 2 Report
In this manuscript, nicotine or/and 3-phenylpyridine imprinted chitosan-based thin films were developed for Quartz Crystal Micro scale (QCM) analysis. A number of discussions were provided along with experimental data. The present study may be of potential use in the fabrication of molecular imprinted polymers. However, the present manuscript seems not be organized well and the novelty has not been fully presented.
- Introduction is general. What is the real novelty of the work? There are numerous papers which describe the application of composite materials for recognition of Nicotine. What are the disadvantages of their methods that overcome in the current work? What is the superiority of the proposed method compare to previous ones? The introduction should revise according to the answer of these questions.
- Performance of MIPs prepared with single template and double template should be compared in depth and propose clear conclusions.
- Results of QCM analysis of nicotine and 3-phenylpyridine should be clearly presented.
- Figures in the manuscript should be clearly presented and reduced to about 6 pieces by merging, and the rest could be moved to Supporting Information.
- Page 14-15, structural schematic diagram of MIP Product A-C should be optimized for scientific and MIP usually refers to the polymer after removal of templates.
- Section 3.7. Instrumentation should be concise.
- Conclusions should be optimized, including summary of the whole work, highlight and put forward some constructive suggestions for the future work.
- Several recently MIPs related publications should be added to strengthen the research background and improve the manuscript organization. For example,
Molecular imprinting: perspectives and applications, CHEMICAL SOCIETY REVIEWS Volume:45 Issue:8 Pages:2137-2211 Published: 2016
Electrochemical sensors based on molecularly imprinted chitosan: A review, TRAC-TRENDS IN ANALYTICAL CHEMISTRY Volume:130, Article Number:115982 Published: 2020
An electrochemical molecularly imprinted sensor based on chitosan capped……, TALANTA Volume: 223, Article Number: 121689, Published: 2021
Voltammetric Sensor Based on Molecularly Imprinted Chitosan-Carbon ……, MATERIALS Volume:13 Issue:3, Article Number:688 Published:2020
................
- It is very necessary to polish the whole manuscript by a native speaker.
Author Response
Thanks for the thorough review, all corrections have been implemented on the manuscript and highlighted yellow.
other comment are in the attachment.

Round 2
Reviewer 2 Report
Reviewers comments have been responded well, I recommend it to be accepted.Author Response
All comment have been attended to and highlighted in yellow
